# Exploratory Systematic Review of Mixed Martial Arts: An Overview of Performance of Importance Factors with over 20,000 Athletes

**DOI:** 10.3390/sports10060080

**Published:** 2022-05-24

**Authors:** João C. A. Bueno, Heloiana Faro, Seth Lenetsky, Aleksandro F. Gonçalves, Stefane B. C. D. Dias, André L. B. Ribeiro, Bruno V. C. da Silva, Carlos A. Cardoso Filho, Bruna M. de Vasconcelos, Júlio C. Serrão, Alexandro Andrade, Tácito P. Souza-Junior, João G. Claudino

**Affiliations:** 1Research Group on Metabolism, Nutrition and Strength Training, Department of Physical Education, Jardim Botânico Campus, Federal University of Paraná, Curitiba 80210-132, PR, Brazil; tacitojr2009@hotmail.com; 2Sciences Center of Health and Sport, Laboratory of Sport and Exercise Psychology, Physical Education Department, State University of Santa, Catarina 88080-350, FLN, Brazil; d2aa@hotmail.com; 3Associate Graduate Program in Physical Education, Federal University of Paraíba, João Pessoa 58059-900, PB, Brazil; heloianafaro@gmail.com; 4Sport Performance Research Institute New Zealand, School of Sport and Recreation, Auckland University of Technology, Auckland 1010, New Zealand; lenetsky@gmail.com; 5Canadian Sport Institute Pacific, Victoria, BC V9E 2C5, Canada; brunamavgnier@hotmail.com; 6Laboratory of Psychophysiology and Performance in Sports & Combats, School of Physical Education and Sport, Federal University of Rio de Janeiro, Rio de Janeiro 21941-599, RDJ, Brazil; aleksandrofg@gmail.com; 7Exercise and Sport Science Laboratory, Keiser University Orlando, Sports Medicine & Fitness Tech/Exercise Science, 5600 Lake Underhill Road Orlando, Florida, FL 32807, USA; sdias@keiseruniversity.edu; 8Department of Physiology and Product Development Limber Software, Balsam 15140-000, SP, Brazil; alberzotiribeiro@gmail.com; 9Department of Physical Education, University of Itaúna, Highway MG 431-Km 45, Itaúna 35680-142, MG, Brazil; brunobjjbh@gmail.com; 10Laboratory of Biomechanics, School of Physical Education and Sport, Campus São Paulo, Universidade de São Paulo, São Paulo 05508-030, SAO, Brazil; carlos.filho@alumni.usp.br (C.A.C.F.); jcserrao@usp.br (J.C.S.); or joaogustavoclaudino@gmail.com (J.G.C.); 11Research and Development Department, LOAD CONTROL, Contagem 32000-000, MG, Brazil; 12Center for Health Sciences, Group of Research, Innovation and Technology Applied to Sport (GSporTech), Department of Physical Education, Federal University of Piauí, Teresina 64000-850, PI, Brazil

**Keywords:** MMA, injuries, weight loss, technical and tactical analysis, psychobiological

## Abstract

This review aimed to analyze the findings in the literature related to Mixed Martial Arts (MMA) through an exploratory systematic review and to present the state of the art from a multifactorial perspective. The review was conducted in accordance with the PRISMA statement, with a search performed in the Scopus, PubMed, and Web of Science databases. Participants were competitive athletes (amateurs or professionals) of regional, national, or international levels. Of the 2763 registries identified, 112 studies met the eligibility criteria. The pooled sample size and age were 20,784 participants, with a mean age of 27.7 ± 6 years for male and 28.9 ± 3 years for female, with the vast majority of athletes being male (94.9%). MMA athletes were 17.2% amateurs, 73.8% professionals, and 9% were not reported. The scientific literature related to MMA reported injuries (*n* = 28), weight loss (*n* = 21), technical and tactical analysis (*n* = 23), physical fitness (*n* = 8), physiological responses and training characteristics (*n* = 13), psychobiological parameters (*n* = 12), and interventions applied to MMA athletes (*n* = 7). Therefore, this exploratory systematic review presents practitioners and researchers with seven broad summaries of each facet of performance of importance in this population of athletes.

## 1. Introduction

Mixed Martial Arts (MMA) is a full contact combat sport characterized by its high degree of freedom in offensive and defensive approaches, resulting in the inclusion of techniques and tactics from multiple combat sport styles [1]. MMA has gained legitimacy and uniformity of rule sets after a tumultuous and unregulated introduction in North America [2,3,4]. Victory in a bout can occur through submission, knockout (rendering an opponent unconscious), technical knockout (rendering an opponent unable to continue fighting), stoppage by the ringside doctor, or a decision by judges. A judges’ decision is based on the quality and effectiveness of the strikes and grappling techniques [5]. Judges’ decisions can result in the following scores: 10–10 round even between fighters, 10–9 round win by a very small margin by one of the fighters, 10–8 round won by a wide margin by one of the fighters, or 10–7 round won overwhelmingly and dominantly by one of the fighters [5]. MMA is one of the fastest growing sports in the world [6]. In 26 years, since its rebranding for Western audiences in 1993, the number of fights per year have increased by more than 100 times. In 1993 there were 72 fights, in 2019 there were 19,877 fights, in 2020 there were only 8884 fights at the expense of the COVID-19 pandemic [7], and in 2021 there were 13,779 fights [7]. As this sport has become more regulated and accepted globally, research into MMA has increased, particularly in the last decade [7,8].

Due to the nature of combat sports, early research in the topic had focused on injury risk in the sport [9,10]. As one of the most explored areas in MMA literature, these studies have established many of the epidemiological factors resulting from participation [11,12]. This research has highlighted neurological, musculoskeletal, and endocrine system risks related to preparation for, and competition in, MMA [12,13,14,15,16]. Scientific literature on athletic performance in MMA emerged in the early 2010s, with researchers examining athlete profiling, time–motion analysis, weight-cutting strategies, psychological factors, and intervention studies [1,14,17,18,19]. The most recent published reviews on MMA cover specific approaches such as muscle injuries [12], magnitude of rapid weight loss and rapid weight gain [20], physical and training characteristics [21], quantification of typical training loads and periodization practices [22], and application of sport psychology [19]. Furthermore, Spanias et al. [23], in a brief review, reported that anthropometric and physiological profiles of MMA athletes may not vary among high-level athletes. Based on the Applied Research Model for the Sport Sciences implementation studies (i.e., last stage: studies to determine how effective at improving performance the previously identified interventions are when used in a real-world scenario) were not found in the MMA literature [22]. Furthermore, we observed that there is a lack of systematic reviews using different databases (e.g., Web of Science, Scopus). As a result of these approaches performed (i.e., brief reviews and narratives), a large difference was found in the number of studies selected, 19 and 70 studies, [22,23] respectively, and additional studies may be found in the scientific literature. To the best of our knowledge, there are still no methodical, comprehensive, transparent, and replicable studies that have summarized the overall perspective of MMA athletes, resulting in an integrated analysis for a better scientific understanding and more efficient practical applications. Therefore, the objective of the present study was to analyze the findings of the scientific literature related to MMA through an exploratory systematic review on the subject and to present the state of the art of the sport from a multifactorial perspective.

## 2. Materials and Methods

We adopted the Preferred Reporting Items for Systematic Reviews and Meta-Analyses (PRISMA) guidelines [24]. The registration was published on the INPLASY website with the registration number: INPLASY202240158 and the DOI number: 10.37766/inplasy2022.4.0158.

### 2.1. Procedure

The selection process and data extraction methods were completed by three authors (A.F.G., S.L. and J.C.A.B). A fourth author (J.G.C.) supported this team and aided them in the methodological process. The quality appraisal was completed by three other authors (A.F.G., C.A.C.F. and A.B.).

### 2.2. Search Strategy

Three electronic databases (PubMed, Scopus, and Web of Science) were systematically searched until January 2022. The command line “Mixed Martial Arts” or “Combat Sport” was used during the electronic search.

### 2.3. Eligibility Criteria

The titles and abstracts were reviewed based on the following inclusion criteria: (1) the study was written in English, (2) the study was published as original research in a peer-reviewed journal as a full text article, (3) data were reported for the MMA athletes, and (4) the participants were competitive athletes (defined as regional, national, and international). After this first screening, the eligibility criteria according to PECO were applied to the remaining full manuscripts. (P)articipants: healthy, MMA athletes of any age, sex and level. (E)xposure: exposure to the MMA training and/or competition. (C)omparators: control groups are acceptable, but not mandatory. (O)utcomes: data reported from the MMA athletes. Based on the title, main objective of the study and according to the main findings of each study, categories were created for analysis. When necessary, the first and last author (JCA and JGC) entered into an agreement to decide if one study could be added in more than one category. For example: study title = “Repeat Effort Performance is Reduced 24 Hours After Acute Dehydration in Mixed Martial Arts Athletes” and study purpose = “This study sought to determine the influence of acute dehydration on physical performance and physiology in mixed martial arts.” Thus, the referred study could be added in the “weight loss” and “physical fitness and performance” categories. However, because of the main findings found (i.e., “Acute dehydration of 4.8% body mass results in reduced physical performance 3 and 24 h after a dehydration protocol”), this study was added just in the “weight loss” category.

### 2.4. Quality Assessment

The quality of all studies was evaluated by three authors (S.L., C.A.C.F., and A.B.) using evaluation criteria (Appendix A: Risk of bias assessment criteria) based on the study by Saw et al. [25], which has been used in previous systematic reviews [26,27]. Scores were allocated based on how well each criterion was met, assuming a maximum possible score of 8 (low risk of bias). Studies with a risk of bias score of 4 or less were considered poor and were excluded. Risk of bias was assessed independently of study appraisal using the GRADE guidelines [28]. This takes into account randomization, concealment of allocation, blinding of outcomes assessment, incomplete outcome data, selective reporting, and other biases, such as stopping early for benefit or the use of non-validated outcome measures.

## 3. Results and Discussion

The initial search returned 2763 articles (Figure 1). After the removal of duplicate articles (*n* = 1835), a total of 1570 studies were retained for full text screening. Following eligibility assessment, 153 studies were excluded as they did not meet the inclusion criteria. Thus, 112 studies were included in this systematic review.

### 3.1. Characteristics of the Studies and Risk of Bias

The grouped sample size was 20,784 participants, with mean age of 27.7 ± 6 years for male athletes and 28.9 ± 3 years for female athletes, with the vast majority being male athletes (94.9%). MMA athletes were amateurs (17.2%), professionals (73.8%), and undefined (9%). All studies included had a low risk of bias, with scores > 4. The scientific literature related to MMA reported studies on the topics of injuries (25%), weight loss (18.8%), technical and tactical analysis (20.5%), physical fitness (7%), physiological responses and training characteristics (11.6%), psychobiological parameters (10.7%), and interventions applied to MMA athletes (6.3%). The mean bias score of the studies was 7 ± 1 (see Appendix A: risk of bias score).

### 3.2. Main Findings

The 112 studies included in the systematic review are described in detail in Appendix A.

#### 3.2.1. Injuries

As MMA includes grappling, striking and throwing techniques, in addition to those that aim to damage joints resulting in bruises, sprains, and even fractures [10]. Due to these inherent characteristics of MMA, studies exploring injuries in the sport have received substantial attention in the scientific literature. In this present study, investigations of injury comprise 25% of all manuscripts. Injuries were reported with different methodologies, such as athlete exposure (AE) [13,29,30], time exposure [29], combat time exposure [30], fight participation [30,31,32,33], and competitor rounds [31,33]. The overall injury prevalence in MMA fighters was 8.5% of fight participations or 5.6% per rounds fought [33], values that are lower than those reported in Olympic combat sports during training (58%) compared to with the competition (42%) [34], still on MMA, 4.1 injuries (95% CI = 3.48 to 4.70) per 100 min of exposure [29], 64.9 injuries per 1000 combat minutes [30], 23.6 to 28.6 injures per fight participations [30,31], 12.5 injuries per 100 competitor rounds [31], and 23.6 (95% CI 20.5 to 27.0) [29] to 51 injuries per 100 AE [13].

There are some inconsistences about the relationship of athlete’s characteristics (i.e., age and weight) and the risk of injuries [30,31]. Some authors suggest that older athletes were associated with a higher risk of injury [30]. However, others showed that age did not statistically increase the likelihood of injuries [31]. When analyzed by weight class, some authors suggested that heavier athletes had the highest injury incidence (75 injuries per 100 AE) [13], while the strawweight athletes had the lowest injury incidence (39 injuries per 100 AE) [13]. Alternatively, others showed that athletes’ weight did not statistically increase the likelihood of injuries [30]. There are inconsistent data about injury incidence and injury rates across genders also. Some studies suggested that the incidence of injuries is higher in men than woman [4,30], with an average of 3.9 injuries per subject in male fighters and 2.3 injuries per subject in female fighters [4], and 54 injuries per 100 athlete exposures in males and 30 injuries per 100 athlete exposure in females [13], while others suggested that there are similar injury rates between men and women [33]. Figure 2 presents the main findings regarding injuries in MMA athletes.

The relationship between competitive level and fight experience has also been investigated in several studies, but inconsistent results were found again [4,30,33]. It is suggested that professional fighters suffer more injuries than amateur fighters [4,33], resulting in three times higher injury rates [4]. Moreover, it has been shown that lower ranked fighters had more than two times higher injury rates than those higher ranked [4]. However, others showed that fight experience did not statistically influence the likelihood of injuries [4]. Most MMA injuries were sustained during competition [35,36]. Regarding fight results, a losing fighter is 2.53 times more likely to be injured than a winning fighter (OR = 2.53, 95% CI 1.88 to 3.42, *p* < 0.01) [30,31].

In summary, injury incidence and injury rates in MMA seem to vary according to the evaluation method. Five differing methods of measuring MMA related injuries were found in the literature. This inconsistency makes it difficult to identify the exact number of injuries among MMA fighters, as well as to compare injury data across different studies or different sports modalities. Therefore, future studies should consider adopting a standard method to assess and report MMA related injuries.

The most common injury locations reported for MMA athletes were the head, face, and neck, followed by upper extremities, trunk, lower extremities, and all other body locations [4,13,29,36,37]. It has been shown that some differences on injury location between men and women exist, with upper limb injuries being more frequent in females [13] and lower limbs injuries being more frequently in males [13].

The higher frequency of head, neck, and face injuries reported seems plausible, and it is likely because attacking the head increases the chance to win by knockout or technical knockout (KO/TKO) [38]. Researchers found that the rate of fights finished by KO was 6.4 per 100 athletes-exposures, by TKO was 9.5 per 100 athletes-exposures, and combined incidence of match-ending head trauma of 15.9 per 100 athletes-exposures, which represents a higher rate when compared to other combat sports [39]. It is well established that head impacts can results in concussion and similar symptoms [29,38,40], especially when MMA athletes are finished via KO/TKO [29,38], with a higher risk of head trauma in heavier fighters [38,41,42]. The chance of a fight stopping by KO/TKO is 80%, 100%, and 206% higher in middleweight, light heavyweight, and heavyweight, respectively, when compared to the lightweight class [38]. 

Regarding injury type, injuries that occur near to the head, such as facial laceration [30,31,43,44], which is one of the most common reported reason of doctor stoppage in MMA (~90% of cases), and concussions [29,37,38,45], were the most common reported. Other types of injuries such as abrasion, contusion, fractures, strains, and sprains were also observed between MMA athletes [4,31,35,36]. Head traumas are a major concern due to the negative effects that this injury leads to, such as headaches, loss of balance, visual impairments, emotional disturbance, and neurocognitive deficits [38,39,46,47,48]. At professional events, such as the UFC, a head injury rate of 35 injuries per 100AE is reported [43]. This value is higher than that proposed by the systematic review of injuries by Thomas and Thomas [12]. A possible explanation for this divergence would be the inclusion of other promotions that demonstrate lower values in injury rates [43]. In addition, male athletes have higher rates of head trauma than women, making it the most observed injury with traumatic brain injury TCE [43]. 

Among MMA fighters, a common practice is to rest for two days after a head injury before returning to the sport or competitive practice. It is suggested to avoid physical exertion until athletes become asymptomatic, as early return can potentially worsen head injuries and increase the chance of developing neuronal diseases such as Alzheimer’s disease and chronic traumatic encephalopathy [40]. However, current data show that after a serious injury, which prevents an athlete from returning to combat, only 94% of fighters tend to return to sport after injury [49]. Additionally, a decrease in performance was observed after injury associated with age. Athletes > 35 years who had suffered injuries are more likely to not return to sport [49].

Regarding brain injuries, a longitudinal study demonstrated temporal trends in structure and cognition—positive and neuropsychiatric outcomes that diverged when investigated by weight category and type of combat sport [48]. This evidence suggests that, although weight category has not been preliminarily identified as a determinant of early brain health among professional athletes, it modifies the course of neurodegenerative transformations over time, as the similar neurological effects of repetitive impacts on the head (RHIs) are concentrated [48]. Currently, few studies have investigated the effects of head accelerations and its outcomes. Lota et al. [50] reported that rotational acceleration (RA) is greater after direct attacks to the head. Consequently, a possible increase in rotational acceleration (RA) is a strong predictor for traumatic brain injury (TBI) [50]. Tiernan et al. [51] reported 541 impacts confirmed over 30 events and found that striking the opponent on the side of the head was associated with concussive injuries [45]. Seeking to improve the prevention of neural damage, whether pre- or post-fight, Matuk et al. [52] explored the feasibility of analysing extracellular vesicles as an analysis tool, finding that the method was able to differentiate more severe injury mechanisms and methods (knockout or technical knockout by punches or kicks) through gene expression [52]. Furthermore, evidence has been found that analysing salivary and serum microRNAs (miRNAs) have been promising in predicting mTBI after head impact in MMA bouts [53]. Finally, other common type of injuries in MMA is ophthalmic injuries caused by lacerations of structures around the eye, commonly resulting in fight-loss by those that receive such injuries [44]. As such, it is recommended that boxing gloves should be used during training, as these reduce the intensities of the impact on the head, minimizing injuries in the facial or eye areas [46].

#### 3.2.2. Weight Loss

Rapid weight loss (RWL) is a common strategy used by MMA athletes to reduce their body mass (BM), aiming to reach the limit of weight division. The process of RWL usually starts one/two weeks before competition, and the athletes lose 2–10% of their BM [54,55]. The prevalence of RWL among MMA athletes is between 88% [56] and 95% [57], with greater predominance among professional athletes compared to amateurs [58]. In contrast to grappling modalities, such as judo, Malliaropoulos et al. [59] reported in their study with British judo athletes a prevalence of RWL of 84%. Moreover, the RWL incidence in MMA is higher than other combat sports [55,57]. Barley et al. [54] reported a reduction of 9.8 ± 7.9 kg of BM, while Santos-Junior et al. [60] reported a reduction of ~10%. Many methods were self-reported by the athletes to induce RWL. For instance, Andreato et al. [56] found the use of diuretics, saunas, and exercise in hot rooms as main methods used to cut weight, while Santos-Junior et al. [60] reported the use of a combination of gradual diet (64.2%), restricting fluid intake (62.6%), and sweat suits (55.9%). Moreover, athletes reported the use of more aggressive methods such as diuretics (~49%) and laxatives (~32%) [60]. Recently, Connor and Egan [55] also reported using water loading and hot clothing alternative baths as a common method to perform RWL by amateur and professional Irish athletes. The frequencies of “always” or “sometimes” in surveys were reported as 90% for water loading and 76% for hot salt baths [61]. Interestingly, Connor et al. [55] compare the use of hot baths with and without salt, but no differences in the magnitude of weight lost as found. Subsequently, Connor and Egan reported no differences between a hot freshwater bath (FWB) and a hot bath with ~1.6% Epsom salt added (SWB) for body mass loss. The authors state that hot baths lasting 2 h are an effective method of losing ~2.0% RWL body mass [62]. Some authors also investigated the source of information to induce RWL. The athletes cited magazines, social networks, coaches, training partners, friends, and physical trainers as sources, although some mentioned a nutritionist service [16,60,61].

The inducement of hypohydration is the primary reported method to provoke RWL and has significant adverse effects, as it can cause psychology [63,64], physical, or health decline [65]. The reduction in bodily fluids resulting from exercising in hot environments, using plastic clothing, laxatives, and diuretics, might generate electrolyte imbalances, especially calcium, which might lead to less bone mineralization and cause stress fractures [66]. Additionally, the use of diuretics produces hypokalemia, which is the reduction in body potassium in the blood, altering the activity of the sodium–potassium pump, which can lead to death [66]. Physiological disturbances on hydration status, hormonal balance, and markers of muscle damage were also reported [14,65,67]. In addition, a decrease on performance [14,18,54,57], emotional and mood disturbances [64], and cognitive impairment [68]. The literature has reported reductions in muscle strength levels, aerobic fitness, reduced plasma and blood volume, decreased myocardial efficiency and maximal oxygen consumption, reductions in blood fluid, renal function and the volume of fluid filtered by the kidneys, reductions in glycogen stores, impairment of thermoregulatory processes, and electrolyte balance [16,54,57,66,67,69,70]. Additionally, RWL is reported to cause physiological damage to mental function by generating deficits in concentration, memory, cognitive processing speed, and an increased risk of developing eating disorders [66,71,72].

The negative influence on physiological, physical performance, or psychological markers due to RWL remains, even after 24 h of recovery [73]. These problems might occur regardless of the time interval between weigh-in and the fight itself, as physiological parameters such as hydration status, salivary nitrate, and energy availability may not be restored enough [3,65,68,74]. Possibly, this behaviour is encouraged by the time between the official weigh-in and the fight (i.e., 12 to 32 h), which allows for weight recovery so that an athlete fights one to two categories above the official weigh-in [3,74].

There is no consensus on the effect of BM recovery on the fight outcome [35]. The act of weight regain (WR) after weigh-in is reported as a common act among wrestlers of both genders with no differences (*p* < 0.005) between relative values of BM gains between genders [75]. Kirk et al. [76] showed that the magnitude of the WR does not predict victory or defeat in a professional cohort of MMA athletes. Alternatively, Coswig et al. [63] found that the magnitude of WG was greater in winners than losers, which was more important than the magnitude of RWL. It is important to highlight that the winners showed higher total caloric intake and absolute intake (g) of carbohydrates, proteins, and lipids during the WG than losers, which might partially explain the result. Faro et al. [77] demonstrated that regardless of gender, fighting category, or competitive level, the percentage of WG can be used as a predictor of competitive success in the fight; every 1% of additional WG increases the probability of competitive victory by 4.5%. Figure 3 presents the main findings regarding weight loss in MMA athletes.

#### 3.2.3. Technical and Tactical Analysis

MMA bouts are characterized by a variety of actions (e.g., grappling, striking, clinching, etc.) that give the sport an intermittent activity characteristic, with reported effort-pause ratio between 1:1 and 1:4 [78,79]. The MMA rule set also influences the characteristic of technical–tactical behaviour, as the athletes choose their combat strategies based on what types of attacks are allowed [80]. Hence, changes in the rules might lead to differences in technical–tactical actions, such as strikes landed and injures [80]. Moreover, age seems to be an essential influent component on technical–tactical actions in MMA. Dos Santos et al. [80] demonstrated a reduction in technical quality after ten years of competitive performance (*n* = 45 professional athletes; age: 34–54 years). These results suggest that technical aspects are susceptible to age.

Technical–tactical and time research was conducted in this review using retrospective data from UFC fights. The most common characteristics observed were the frequency of spatiotemporal work (keeping distance, clinch, groundwork, etc.), motor actions (strikes attempted and landed, submissions, chocks, locks attempted, etc.), technical–tactical actions (preparatory actions and fighting in both standing and groundwork), and effort-pause (i.e., high and low intensities in both standing and groundwork), following a previously established protocol [78,81,82,83,84,85,86]. Miarka and colleagues analysed these data through comparison between weight divisions [78,86], sexes [87], fight outcomes [82,87], type of fight outcome [87,88], actions during the rounds [84], and home advantage [85], using technical–tactical, time–motion, and/or motor action analysis as an outcome. The aim and main findings of these studies are summarized in Appendix A. Furthermore, Antoniêtto et al. [89] compared the pacing strategy and technical–tactical actions during individual rounds (first, second, and/or third rounds). The analysis identified a shorter duration of “standing preparatory activity”, groundwork, and fewer attempted and landed strikes to all parts of the body in the first round of fights which also ended in the first round. Moreover, a similar pattern was found in the second round of fights that ended in the second or third rounds [89]. Additionally, reduced groundwork was found in the third round of fights that ended in the third round. The authors argue that this information might help the coaches strategize pacing strategies in training and consequently improve performance. Likewise, these pacing strategies could be used to distribute offensive and defensive efforts to efficiently distribute the athlete’s exertion, in respect to their individual strengths and weaknesses [88,89]. Finally, Miarka et al. [90] calculated how spatiotemporal and technical–tactical actions increased a fighter’s probability of winning. The authors found that “keeping distance”, “head strike landed”, “offensive passes”, and “effective takedowns” were the most significant actions related to the probability of victory [76]. More recently, an investigation based on more than 900 successful submissions by UFC athletes found that 15.5% of the fights ended up by strangulation, with the rear naked choke used in 49.1% of those fight finishes, and that 76.2 % of total submissions were used by grappling-style fighters [91,92].

The aforementioned characteristics are important for the training process, providing insight into opponent’s potential actions, identifying winners’ frequent actions, sex differences, and weight category characteristics [84,87,88,89]. For example, females who won their fights by KO/TKO/Submission presented a high number of attacking and grappling actions, while those who won by decision had a high number of standing strike actions [82]. These differences were observed at a rate of ~0.5 strike attempts per second of fight in the first round and ~1.0 strike attempts per second of fight in the second and third rounds [93]. Alternatively, male winners maintained a high frequency of actions and increased the number of strikes to the head and body throughout all rounds, with a high frequency of takedowns in fights that end in a judge’s decision [86,88,89,94] (Table 1).

Figure 4 presents the main finding regarding technical and tactical analysis in MMA athletes.

#### 3.2.4. Physical Fitness and Performance

Given the growth of MMA, a greater understanding of the attributes possessed by fighters is needed to inform future research and training. Somatotype and body composition results indicate that MMA athletes have a predominantly mesomorph body type (6.4 ± 0.8) [97], with variation in body fat percentage ranging from 8.5 to 14.9 ± 7.2 [21]. Furthermore, the average values of body fat percentage observed were 13.4 ± 5.6. These somatotype and fat percentage values were similar to those found in other combat sports [23].

Cardiorespiratory endurance, expressed by maximal oxygen uptake (VO_2max_), has been found to range in MMA athletes from 44.2 ± 6.7 to 62.8 ± 4.9 mL/min/kg^−1^ in testing using the lower body and from 45.8 ± 2.9 to 55.0 mL/min/kg^−1^ in testing of the upper body, with mean of 45.5 mL/min/kg^−1^ [23] (Figure 5). These values were higher than the general population, equivalent to other combat sports, but lower than those seen in traditional endurance athletes [23]. MMA bouts most commonly have three to five rounds consisting of three or five minutes each, with one minute of rest in between each round [98]. Under these circumstances, the active energy system is primarily oxidative metabolism [82]. Therefore, the development of the aerobic system is key for MMA athletes maintain intensity throughout the fight and to accelerate recovery between high intensity actions [21].

While the physiological demands of MMA are primarily derived from the aerobic energy system, anaerobic metabolism still supplies key energetic demands during commonly occurring high intensity efforts [98]. In the literature, the most commonly used tool to measure anaerobic power and capacity is the Wingate test. Reported absolute and relative values of peak and mean power, as well as fatigue index results in MMA athletes, were similar to athletes of other combat sports [99].

Neuromuscular testing of MMA athletes has demonstrated intriguing and contradictory results. Marinho et al. [97] reported poor performance on the horizontal jump and maximum strength tests, while James et al. [15] reported good performance for 1RM squat and vertical jump. A possible explanation for this discrepancy between the results is the difference in the test used to analyse lower body power (i.e., horizontal. in Marinho et al. [97] and vertical jumps in James et al. [98]), and the data collection period (i.e., preparatory period in Marinho et al. [97] and not described in James et al. [98]). Despite the contradiction in the literature, lower-body neuromuscular capabilities have been found to distinguish between the high- and low-level MMA competitor, specifically using 1RM squat strength, reactive strength index (RSI), and peak power, force, and velocity in incrementally loaded jump tests [15]. Likewise, comparisons between semi-professionals and amateur MMA athletes found that sprint performance and repeated sprint ability were higher in semi-professionals than amateurs [98]. Comparisons of countermovement jump kinetics further identified that differences between amateur and professional athletes could be identified by higher values in power and RSI and lower eccentric time and impulse, suggesting a higher lower-body neuromuscular control in the higher levels in MMA [99,100]. Thus, the performance on neuromuscular tests might help distinguish the level of MMA athletes, an important factor to be considered during strength and conditioning programming. Figure 5 presents the main findings regarding physical fitness and performance in MMA athletes.

#### 3.2.5. Physiological Responses and Training Characteristics

The intermittent nature of MMA fights induces high metabolic demand [101,102], reflected by an increase in blood lactate (LAC) after training (0.00197 mmol·L^−1^) and official matches (0.00272.7 mmol·L^−1^) [103]. De Souza et al. [104] report similar values between winners and losers during different time periods: −24 h pre-competition (0.00132 ± 0.00047 and 0.00144 ± 0.00027 mmol·L^−1^), 1 h (0.00202 ± 0.00061 and 0.00222 ± 000042 mmol·L^−1^), 0 h (0.00237 ± 0.00318 and 0.00388 ± 0.0013 mmol·L^−1^), and 24 h after competition (0.00172 ± 0.00068 and 0.00182 ± 000047 mmol·L^−1^).

A significant effect was observed in some phases (−24 h to +24 h) between hormonal and biochemical markers. In the hormonal profiles, the behaviour of testosterone (T), cortisol (C), and the testosterone/cortisol ratio (T/C) were observed with a decrease in the T and T/C ratio regardless of performance (gain/defeat) between moments. Initials were (−24 h to 0h; and −24 h to −1 h and −1 h to 0 h) (see Appendix A) [104]. At a later time, increases (*p* < 0.0001) occurred between 0 h and +24 h for T and T/C (see Table 2) [104].

When checking the variables C, LAC, and GLU, De Souza et al. [104] report a contradictory performance, with their values increasing significantly (*p* < 0.0001) between moments −24 h and 0 h and from −24 h at −1 h and −1 h at 0 h pre-fight, C, LAC, and GLU, showing a significant reduction behaviour (*p* < 0.0001) between the moments 0 h and +24 h pre-fight, C, LAC, and GLU, respectively). For the variable CK, significant reductions (*p* < 0.0001) are shown for its serum values between moments −24 h and −1 h pre-fight, as well as a significant increase (*p* < 0.0001) between moments −1 h and 0 h and 0 h and + 24 h pre-fight (see Table 3) [104].

Increased muscle damage markers (e.g., creatine kinase (CK: U/L)) reports median and interquartile range (IQR) of athletes not employing rapid weight loss procedures NWL and athletes employing rapid weight loss procedures (RWL) during Pre-match and Post-match. NWL, CK pre-match with median 230 IQR (136–345.1), post-match with median 252 IQR (146.1–407.1) and RWL, CK pre-match with median 259 IQR (165.5–2130) and median 306 IQR (200–2077.5) [65]. Coswig et al. [102] report on a comparison of a sample of fighters during simulated sparring matches versus real combat, in which they suggest that there are no differences between groups in blood lactate values Official: Pre-competition = 4 (3.4–4.4) mmol/L, Post-competition = 16.9 (13.8–23.5) mmol/L; Simulated: Pre-competition = 3.8 (2.8–5.5) mmol/L, Post-competition = 16.8 (12.3–19.2) mmol/L; *p* < 0.001). From a practical point of view, these data suggest that sparring training, when performed at high intensity, produces the same effects as real combat, as it generates similar damage to biochemical markers of muscle damage. Conversely, data from Ghoul et al. [105] report CK values (U/L) in simulated competition after Trest (Trest): 432 ± 606, round 1 (Trd1), round 2 (Trd2) and round 3 (Trd3) 505 ± 668, then 30 min (Trecovery30min) 468 ± 566 and 24 h (Trecovery24h) 598 ± 451, post simulated competition. Wiechmann et al. [106] demonstrated that during official fights there is damage (*p* < 0.001) in the musculoskeletal system due to the exchange of blows to the lower and upper limbs of the body. Likewise, eccentric actions during the act of deceleration during kicks are associated with muscle damage markers [106]. The enzyme creatine kinase (CK) and the concentration of myoglobin (Mb) showed peak values at 24 h (829 ± 753 U/L^−1^), and Mb reached its peak at 2 h (210 ± 122 µg/L^−1^) after fights [106]. According to the authors, it is suggested that only 20% of the CK peak cannot be explained by the number of hits obtained in the upper and lower body (LBH) and by the number of punches and vertical kicks performed (UKF), while only 13% of the Mb variation cannot be explained by TFD and LBH [106].

Regarding the lactate values, Amtmann et al. [103] reported, in a sample of six athletes, values during training periods ranging from 8.1 to 19.7 mmol·L^−1^ and values after combat ranging from 10.2 to 20.7 mmol·L^−1^ (values measured 2 min after training phase or after combat). Corroborating the data, Kirk et al. [107] reported a mean post-combat lactate of 9.25 ± 2.96 mmol·L^−1^. Furthermore, the authors additionally reported values of media lactate (mmol·L), maximum and minimum values (mmol·L) at different times: (1) At rest (2.7 ± 1.46) max 4.90 min 1.20; (2) After warm up (5.43 ± 2.74) max 8.90 min 2.2; (3) End of Round 1 (6.6 ± 1.99) max 9.20 min 3.60; (4) End of Round 2 (8.67 ± 2.86])max 11.60 min 5.40; (5) End of Round 3 (9.25 ± 2.96) max 13.10 min 5; (6) 5 min post out (7 ± 2.09) max 10.60 min 5.20. Subsequently, Coswig et al. [65] and [102] report that there are no differences between the values between simulated fights versus real combat regardless of whether there is rapid weight loss (RWL) or not (non-RWL) among professional athletes. Lactate (mmol/L) (median (interquartile range); pre to post-fight), Official (pre and post-fight) = 4 (3.4–4.4) to 16.9 (13.8–23.5) mmol/L and simulated (pre and post-fight) fight)= 3.8 (2.8–5.5) to 16.8 (12.3–19.2) mmol/L (2015) and lactate (mmol /L) (median (interquartile range); pre to post-fight), NWL= 4.0 (3.3–4.3) to 16.9 (12.7–24.1) (mmol/L) and RWL= 2.2 (1.7–2.4) to 15.9 (12.1–20) (mmol/L) [65].

These physiological results appear to occur regardless of the outcome of the fight [104] and may persist 24 h after a bout [104,105]. However, muscle damage seems to be related to the total duration of the fight and the number of blows received, especially in the lower body [106]. Regarding the inflammatory responses, professional MMA fighters present markers immediately after the fight, with results in the literature finding mild levels of oxidative stress, possibly mediated by IL-6 and by the defensive responses of thiol-tripeptide glutathione (GSH)-dependent antioxidant agents in blood plasma [108].

Similar to the metabolic responses, MMA participation causes intense perceptual demands, with values of 19 AU in a 6–20 Borg scale, after both training and official matches [103]. In addition, before official matches, glycolytic availability was raised, which may be associated with an effect of psychological stress [102].

Several studies have focused on describing the training characteristics of MMA athletes. Amtmann et al. [9] found that regional athletes performed specific training 3–12 times per week, with almost all athletes supplementing their routine with strength training 1–7 days per week, and some of them self-reporting the use of anabolic-androgenic steroids. According to Fares et al. [109], the use of band ergogenic aid by MMA athletes in the UFC is common. According to Fares et al. [109], doping is present among UFC MMA athletes with an average age of 32 years, in which the weight category was the most prevalent with the highest rate of fighters and anti-doping rule violations (ADRVs): 19.3 per 1000 tests. Between late 2015 and late 2019, *n*= ~1070 UFC athletes were tested 2624 times reporting 102 ADRVs and, of these, *n* = 93 fighters (8.7%) belonged to all weight categories [109]. This yields a corresponding rate of 16.55 per 1000 tests and an ADRVs rate of 8.08 per 1000 tests [109]. Kirk et al. [107], in an innovative and promising study exploring external loading during sparring training through the use of a tri-axial accelerometer, found the average accumulated load of the fighters (PLdACC was of 224.32 ± 26.59 au) and average accumulated load per minute (PLdACC.min-1 was 14.91 ± 1.78 au). These values were reported as greater than other sports, such as netball and soccer. In addition, the researchers reported a post-session lactate mean of 9.25 ± 2.96 mmol·L^−1^ [107]. Furthermore, the same study reported a work/rest ratio (W:R) of 1:1.01. Figure 6 presents the main findings regarding physiological responses and training characteristics in MMA matches.

#### 3.2.6. Psychobiological Parameters

For competitive success to occur in a fight, fighters must present high levels of physical conditioning such as strength, power, agility, and flexibility [21,110,111] and present good sport related mental health. Andrade et al. [19] identified the fundamental role of mental skills throughout a fight, as it is associated with winning athletes (low anxiety, increased confidence, greater ability to control emotions). Sport psychology in the MMA context is of paramount importance in helping to understand feelings and confrontations, whether in life or in moments during combat [19]. Professional athletes show positive changes in managing emotions and performance while also demonstrating higher scores in mental strength when compared to amateurs [19].

Many psychological characteristics of fighters have been investigated and taken into account during the training process [112]. For example, the assessment of mental resistance in amateurs, semi-professionals, and professionals male MMA athletes (27.1 ± 4.8 years) showed that professional athletes demonstrate a better profile of “confidence”, “cognition positive”, “determination”, and greater “mental toughness”, up to 10% higher than amateur and semi-professional athletes [112]. The Positive Cognition (PPI-A) test demonstrated that professional athletes have on average 9.8% higher Mental Toughness scores than semi-professional competitors and 9.3% higher than amateur athletes. For Confidence (SMTQ), professional athletes have on average 9.1% and 7.1% higher scores than amateurs and semi-professionals athletes, respectively [112]. Qualitative investigation found that the self-regulation of internal and external factors is one of the psychological keys during the training process [113]. Factors as motivation, ongoing evaluation, maintenance of routine, self-induction of pain, self-efficacy, and production of stress/fatigue were listed as present in the MMA athlete routine [113]. This evidence suggests that MMA athletes need to pay more attention during the initial phases (amateur and semi-professional) to the psychological characteristics necessary for this combat sport modality [19,112]. Moreover, coping strategies have been very useful in sport dynamics due to the necessity to deal with many challenges during an athletic routine.

The MMA fighter who wants to achieve competitive success will have to undergo high training loads and adapt to them physically, physiologically, and psychologically. Knowing the difficulty and stress in the athletic training process and being able to identify talent and resiliency before its maturation process could help coaches in the choice. Thus, Niewczas et al. [114] reported that there is an interaction between DRD2 rs1799732 dopamine polymorphism and athletic level in the face of knowledge of loss containment and cooperation. According to the authors, a proper understanding of these associations could suggest which athletes would be more suited to combat sports and to facing the imminent risks in the profession [114].

Aggressiveness is also an object of study in MMA athletes’ psycho-social behaviour. According to Little et al. [115], facial cues predicted winners and losers in MMA fights. Independent groups of participants judged fighter’s faces to identify who would win in a physical fight. Observers, unfamiliar with the outcome, were presented with image pairs and asked to choose which of the two males was more likely to win if they fought, while other observers chose between the faces based on masculinity, strength, perceived aggressiveness, and attractiveness [115]. The authors found that individuals performed at rates above chance in correctly selecting the winner as more likely to win the fight than the loser. Moreover, the winners were seen to be more masculine, more aggressive, and stronger than losers. However, the effect sizes for each of these relationships were generally small, which lead to being cautious in the practical applications based on these findings [115]. Otherwise, the presence of a beard, despite being socially judged as a symbol of aggressiveness, masculinity, facial protection, and sexual loftiness, was not related to the fight outcome, being used for intimidation more than face protection [116]. Moreover, the stance predominance (i.e., southpaw, orthodox) and past experience also do not influence fight outcome, but the wingspan did [116]. Lastly, another behavioural action that might appear to be more aggressive and useful during combat is grunting. Indeed, Sinnett and Kingstone [117] found that the grunting increases the force during a kick and could distract the opponent, leading to error and slow response time. Moreover, lower voice tones are associated with successful mating and the reproductive act [117]. As such, a group of researchers used voice samples of 475 MMA fighters in order to predict competitive success. After investigating the relationship between tone of voice and competitive success, the hypothesis was not confirmed, as only 3–8% of the variance can be explained using these variables [118]. Corroborating the data, Schild et al. [119], in their sample of 135 MMA fighters, reported no associations with tone of voice and competitive success.

Psychological characteristics for developing excellence in MMA athletes were reported by Ruiz-Barquín et al. [120]. Amateur, semi-professional, and professional MMA athletes answered a 14-item questionnaire related to the characteristics of the development of excellence. Their responses were analysed and compared to other sports. MMA fighters obtained high scores in several factors, such as: “support for long-term success”, “imagery use during practice and competition”, and “support from others to compete to my potential” [120]. However, the factor “coping with performance and developmental pressures” was worse than other sports [120]. It is possible to conclude that MMA fighters have a better perception of positive factors related to excellence but may need specific psychological training related to stress coping strategies [120].

It has been established that the consumption of ergogenic aids (including banned substances) are commonly used in MMA to improve performance [109]. This use likely stems from their training environment and the professionals who make up the technical committee “reference group” (GR), that is coaches, teammates/training partners, and doctors [121]. According to Petrou et al. [121], this permissible environment and positive perceptions related to doping with banned substances stems from the GR, which makes the athlete feel more confident in their indiscriminate use of performance enhancing substances. Thus, it is relevant that MMA teams to staff a psychologist on their team to adequately analyse injunctive social norms that are misinterpreted or misrepresented, as these are strongly associated with permissive relationships to doping.

Beyond psychological investigations, investigations of physiological/biological aspects influence on perceptual/cognitive/behavioural responses have been investigated in the MMA related literature. Cherepkova et al. [122] compared genetic characteristics (i.e., variable number tandem repeat polymorphism of the dopamine 4 receptor and dopamine transporter) among males with antisocial behaviours (i.e., criminal males), MMA athletes with no history of antisocial behaviour, and the general population. The researchers reported a higher frequency of the D allele among convicts and MMA fighters than among the general population [122]. The tendency of MMA fighters to present violent behaviours is common within the sport, and this behaviour is seen as an attitude related to violence [122].

Pavelka et al. [99] investigated whether neuromuscular fatigue can impair cognitive performance. The authors found that upper-body neuromuscular fatigue decreased reaction time by 1.5% and consistency of response by 15%, concluding that the accumulated fatigue affects cognitive performance [99]. Finally, the quality and quantity of sleep is also an interesting variable of analysis on MMA athletes’ performance. An observational study carried out for six weeks in competitive practice aimed to verify the relationship between sleep and physical performance. The study demonstrated a negative relationship with significant findings between sleep latency (i.e., time to fall asleep after going to bed) and aerobic capacity (i.e., VO_2max_), heart rate recovery, vertical jump, and missed practice sessions [123]. Moreover, sleep latency was also significantly negatively related to heart rate recovery and missed practice sessions [123]. MMA athletes who exhibited consistency in sleep demonstrated stronger relationships with performance tests during the fight prep period [123]. Sleep duration and latency are among the most used sleep quality parameters by individual sports athletes [124]. Despite the findings, only eight athletes were included in the study and only a very limited period of time was investigated. Figure 7 presents the main finding regarding the psychobiological parameters of MMA athletes.

#### 3.2.7. Intervention Studies

The strength and conditioning training programs for the Specific Training group can improve the physical fitness, balance, endurance, and flexibility in professional MMA athletes [125,126,127]. Kostikiadis et al. [127] compared a 4-week intervention of strength training and specific conditioning for short-term, high-intensity, low-volume MMA composed of two sessions, with session (1) (Strength training + Power exercises) and (2) Power exercises + SIT speed exercises) in the performance of national level MMA athletes. The authors reported improvement in performance in aerobic capacity (estimated VO_2max_), average power during 2000 m rowing (−3.4 ± 2.9), power (in SJ (3.0 ± 1.3 cm; 6.1 ± 5.3 W), CMJ height (7.4 ± 4.4 cm; 3.59 ± 7 W), medicine ball throw velocity MBT (right arms: 10.8 ± 4.4 m*s^−1^; left arms: 5.7 ± 7.5 m*s^−1^), 10 m sprint (−3.7 ± 1.4 s) and 2 m take down speed (−22.0 ±4.9 s). Back squat (19.5 ± 10.4 kg), 1RM (bench press (16.1 ± 7.8 kg), deadlift (20.1 ± 10.7 kg), and increase in fat-free mass (−4.5 ± 2.04%; − 4.3 ± 2.3 kg; 3.9 ± 2.5 fat free mass kg) were found only in the specific training group (3.7 to 22.2%; *p* < 0.05; Hedge’s g = −0.42–4.1) [127]. These findings suggest that programs based on strength and conditioning training implementing high intensity and low volume organized according to the specifics of combat demands can generate positive effects on the athletes [127]. Corroborating these results, Tota et al. [126] found that 14 weeks of periodized strength and conditioning with general, specific, and pre-competitive mesocycles proved to be useful, as the training protocol was able to reduce body fat and improve anaerobic peak power (Wingate test for upper limbs) and aerobic capacity (Vo2 and determination of the second ventilatory threshold) in MMA athletes. Chernozub et al. [128,129] conducted two relevant intervention studies with MMA athletes focusing on strength training. The authors reported the importance of strength training models adapted to the predominance of the fighting style (striking or grappling) [129]. Striking athletes used targeted interventions in a high intensity regime, predominantly focused on alactic and lactic energy sources, while the grappling group focused their training on lower intensity regimes with the predominance of the glycolytic energy source [129]. The second study by Chernozub and colleagues focused on MMA athletes who primarily used grappling. A 3-month intervention was performed by two groups separated into: (A) low-intensity, high-volume loading and (B) high-intensity, low-volume loading [128]. Both groups increased their muscle circumferences, group A: 1.9% and group B: 5.5% after 3 months of training [128]. However, although physiological adaptations were different between groups, changes over time were small [128]. In view of the biochemical results of the modifications of the testosterone hormones and the lactate dehydrogenase enzyme in both groups, it validates the use of high intensity loads that favour the evolution of the adaptive capacities of the fighters’ body to perform MMA training [128]. Thus, the need for correction in load parameters (kg) during strength training based on adaptive changes in body composition throughout the training cycle is recommended [128].

Bodden et al. [130] performed an 8-week corrective exercise program aimed at improving functional movement screen (FMS) scores and potentially decreasing injury risk. The authors suggest that the FMS would be a good tool to identify movement disorders and that, after the 4-week intervention, they would improve FMS scores [130], consequently reducing asymmetries at week 4 and 8 compared to baseline values. The authors point out that due to the high ratio of FMS tests and exercises used in MMA training programs, their applicability would be easily understood by fighters [130]. Ignatjeva et al. [131] proposed a method to evaluate or improve the performance of fighters through motor tests. According to the authors, with the progressive use of external resistance using the Keizer Leg Press A421 equipment, it is possible to measure the differences in the reaction time of the lower limbs, whether posterior or anterior.

The use of caffeine was investigated with the aim of improving punch performance (number of punches, strength of each punch, maximum strength, average strength, and frequency or “number of punches per series”) and the perceptual variables. Interestingly, no difference was found when compared to placebo at 5 mg·kg^−1^ of caffeine consumption [132]. Lymphatic drainage methods (manual lymphatic drainage, body flow therapy, and deep swing lymphatic drainage) have been tested to improve post-exercise regeneration in forearm muscles [133]. Zebrowska et al. [133] found that, in MMA athletes, all types of lymphatic drainage recovery interventions (with the use of manual techniques or the use of electrical stimulation) improved the regeneration of the forearm muscles after exercise, as measured by Doppler ultrasound velocity “Ultrasonograf”, and decreased the tension (measured by a myotonometric probe) in each area on the AUC curve (quotient between the amount of tissue deformation and the force applied by the myotonometer times the tissue displacement value at which AUC growth leads to a reduction in surface tension of tissues). The authors demonstrated effects on post-exercise AUC across the investigated sample at all times of recovery (Rec), −20 min, 24 h, 48 h after interventions, and recommend its use [133].

Assessments that allow the identification of changes in movement patterns which may cause functional limitations and interfere with physical performance through screening tools (history of injuries, Time–motion, FMS, etc.) are common methods to reduce the likelihood of injury risk. Moreover, it has been suggested that monitoring fatigue and workload can improve performance and further reduce injury risk. Thus, Kirk et al. [134] reported variations resulting from training loads over an 8-week cycle among 14 athletes, 7 in a competitive cycle and 7 in a general phase. The authors found that training duration (weekly mean range = 3.9–5.3 h), sRPE (weekly mean range = 1287–1791 AU), strain (weekly mean range = 1143–1819 AU), monotony (weekly mean range = 0.63–0.83 AU), fatigue (weekly mean range = 16–20 AU), and soreness did not change within or between weeks [134]. The reported data showed no differences between most investigation variables between groups (competitors vs. non-competitors), the only exception being in the pre-competition week, with abrupt changes in total training time [134]. While the present study demonstrates a possible classification of training intensities, it is necessary to observe the data with caution, given the size of the sample and the level of experience of the fighters. Figure 8 presents the main finding regarding interventions applied in MMA athletes.

## 4. Conclusions

This review exploring MMA, whether in amateur or professional athletes, was based on 20,784 athletes and focused on investigations that evaluated injuries (mainly effects of head injuries: short and long term), effects of weight loss (prevalence 88% to 95% of all participants), technical and tactical analysis (time–motion analysis using winners and losers), physical fitness (reporting the importance of high strength levels and aerobic fitness), physiological responses and characteristics training (blood lactate increases after training (1.97–2.70 nmol/L) and RPE rating from 15 to 19), and psychobiological parameters (“controlled aggression” and sleep disturbances).

Currently, the scientific literature consists of a body of studies on time–motion analysis that allowed an advance in the knowledge area of technical and tactical coaching. However, there is a lack of intervention studies found, which suggests the need for more controlled trials to improve the level of evidence and understanding of sport. Furthermore, according to the inclusion criteria, only manuscripts written in English were selected, and this can be considered a limitation of the present study.

The results of this, the first exploratory systematic review on the subject, highlight the broad, but often superficial, research found in the literature related to MMA. By conducting a methodical, comprehensive, transparent, and replicable survey of the literature, we identified a number of trends in the literature on the subject. Of note, there is a paucity of research on female MMA fighters in multiple areas of study (weight loss, weight regain, nutritional profile, psychological profile, physical profile) and intervention studies that demonstrate a more comprehensive profile of MMA athletes.

Currently, topics including time–motion, weight loss, and incidence and prevalence of injuries have been explored in many published studies and serve to provide a better understanding of sport. Moving forward, this manuscript serves to educate practitioners on current evidence and inform researchers about gaps in the literature.

## Figures and Tables

**Figure 1 sports-10-00080-f001:**
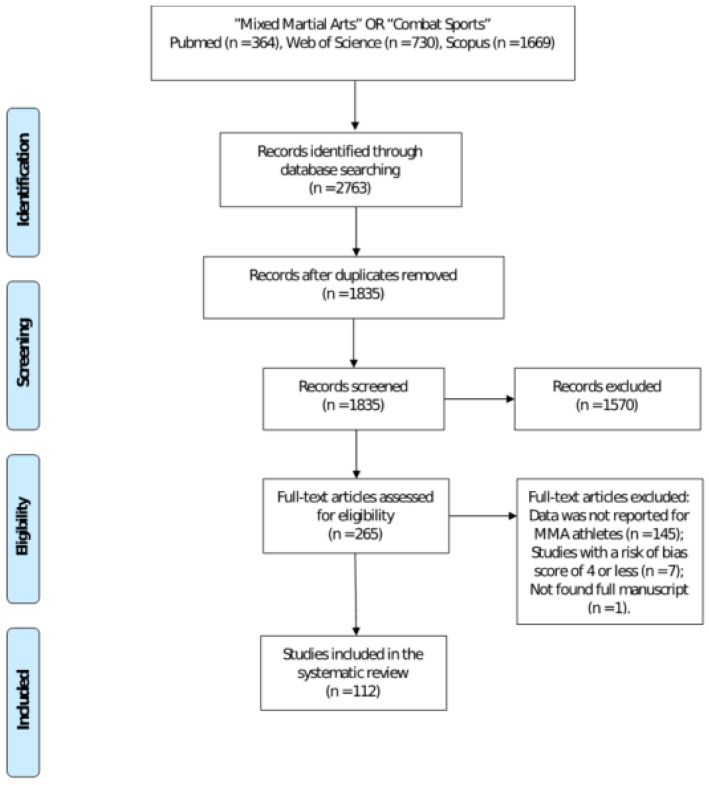
Study selection PRISMA flow diagram.

**Figure 2 sports-10-00080-f002:**
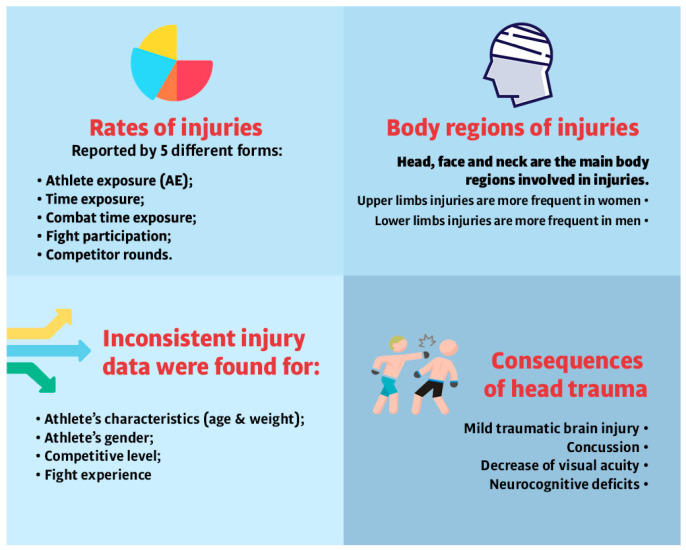
Main findings regarding injuries in MMA athletes.

**Figure 3 sports-10-00080-f003:**
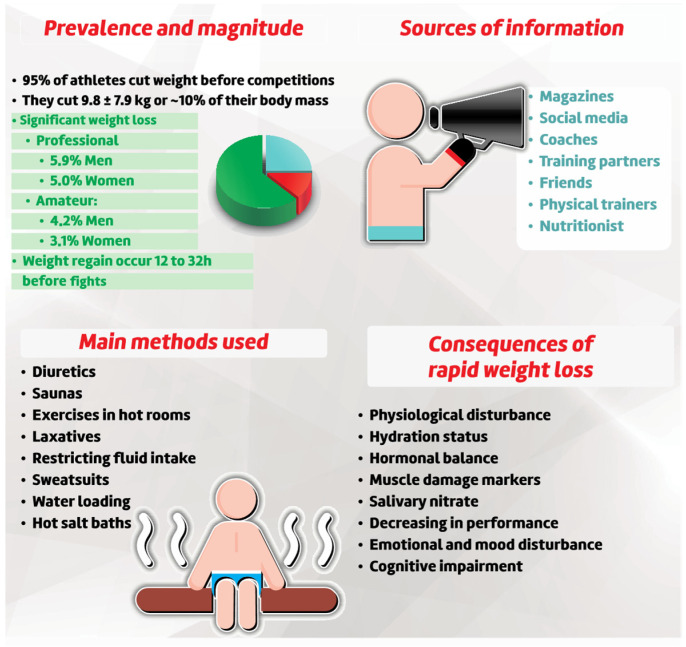
Main findings regarding weight loss in MMA Athletes.

**Figure 4 sports-10-00080-f004:**
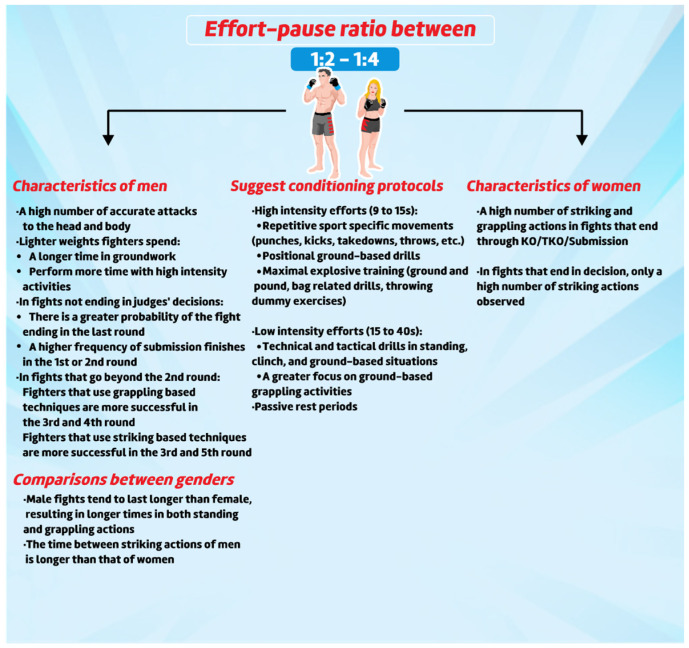
Main findings regarding technical and tactical analysis in MMA Athletes.

**Figure 5 sports-10-00080-f005:**
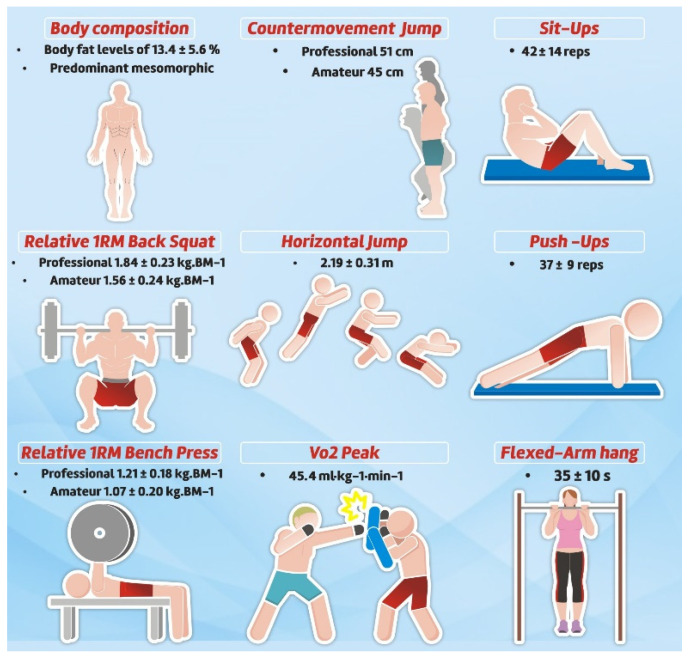
Main findings regarding physical fitness and performance in MMA Athletes.

**Figure 6 sports-10-00080-f006:**
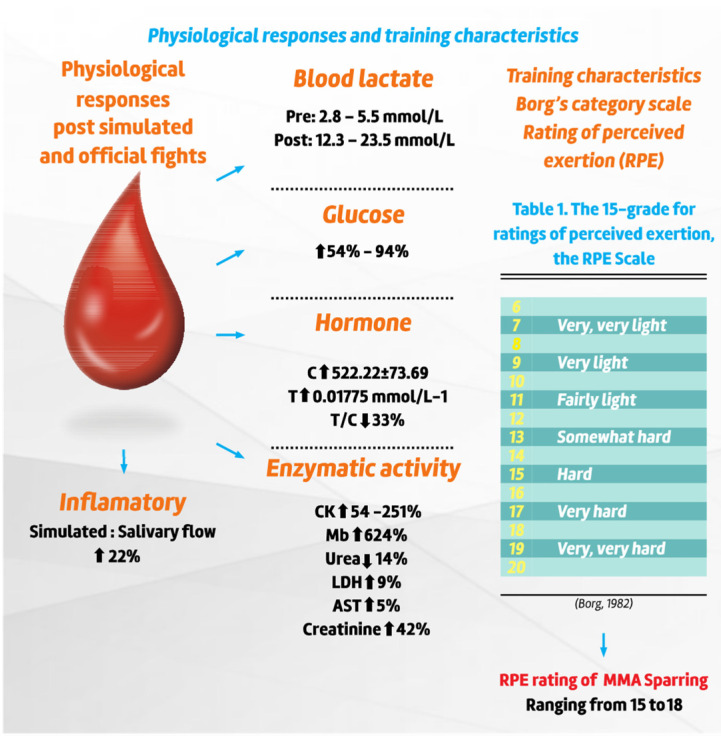
Main findings regarding physiological responses and training characteristics in MMA athletes.

**Figure 7 sports-10-00080-f007:**
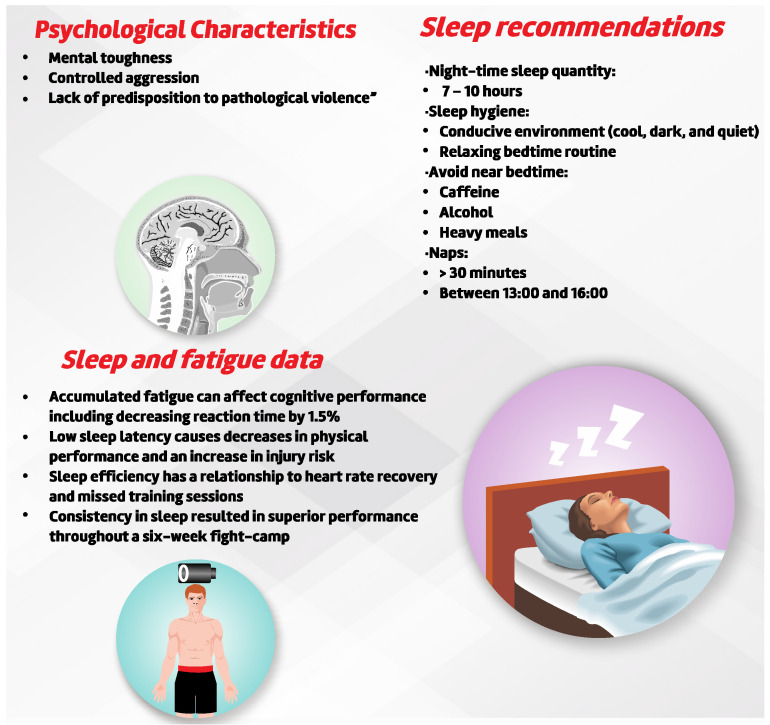
Main findings regarding psychobiological parameters in MMA Athletes.

**Figure 8 sports-10-00080-f008:**
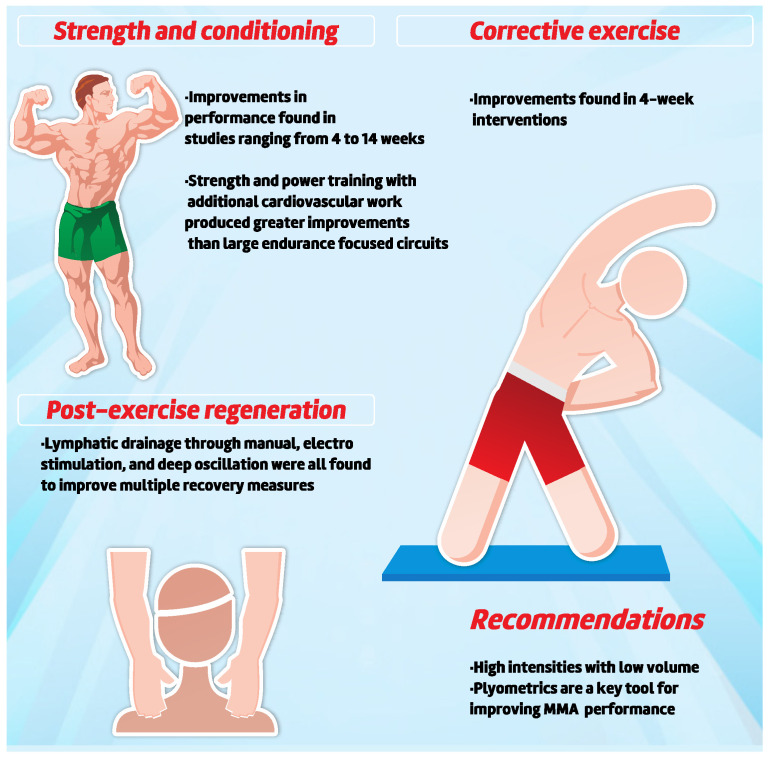
Main findings regarding interventions applied in MMA Athletes.

**Table 1 sports-10-00080-t001:** Compare technical–tactical, time–motion, motor action, and spatiotemporal characteristics among weight divisions, sex, and fight outcomes.

Study	Objective	Main Findings
Miarka et al. [78]	Comparison of time–motion between male weight divisions and rounds in 2097 bouts of UFC events.	Heavyweight athletes presented a shorter high effort time in the first and third rounds. Fly- and lightweight divisions also spent more time in low-intensity in the first round. In the second round, bantamweight spent a shorter time performing lower-intensity actions than fly- and lightweight division. Featherweight spent more time performing high-intensity groundwork than fly-, light-, and middleweight divisions. In the third round, bantamweight spent more time executing groundwork at high intensity than the four other divisions.
Miarka et al. [84]	Comparison of time–motion and technical–tactical analysis between sexes and weight categories (light- and middleweight) in UFC fights.	Male middleweight athletes spent a longer time in standing activity (striking or standing in preparatory activity) than other sex/divisions and longer time on groundwork than male lightweight. Male lightweight presented longer standing fight activity than the other sex/divisions and spend more time between actions in both standing and on groundwork when compared to female middleweight.
Miarka et al. [82]	Comparison of time–motion and technical–tactical analysis between rounds and between winners and losers.	Winners presented higher total strikes and takedown attempts in the first round. Furthermore, the winners performed a higher total of head and body strike attempts in the first and second rounds than in the third round. Submission attempts were more frequent in the first round, while advances in the mount and half-guard were more frequent in the second round. Regarding combat phases, in the third quartile of the first round, the time spent in low-intensity activity was higher than in the third round. In the third quartile of the third round, the time spend on low-intensity activity was higher than in the second round.
Brito et al. [95]	Comparison of the motor actions of winners and losers with consideration of doping status.	Athletes who tested positive for doping had a higher physical performance, such as effort and pause time, however, doping did not reflect better technical performance.
Miarka et al. [82]	Comparison of-motion and technical–tactical analysis between rounds of female combats in the UFC.	The total strike attempts, head strike attempts while standing, and total leg strike attempts while standing were higher in bouts finished by judges’ decision. Total body strike attempts were more frequent in fights ended by split decision. Total head strike attempts in groundwork position and advances to half guard were higher in bouts finished by KO/TKO than by decision. Passes to side control were higher in fights ended vis submission than decision. Advance to mount was higher in bouts finished by unanimous decision than by KO/TKO. Submission attempts were greater in bouts finished by submission than all others.
Dal Bello et al. [88]	Comparison of grappling actions to fight outcomes.	Takedown attempts were more frequent in fights finished by judges’ decision than by KO/TKO. Takedown success was more frequent in decision and KO/TKO finished bouts than those finished by submission. Total submission attempts were more frequent in decision and KO/TKO than in bouts won by submission. All type choke attempts were more frequent in bouts finished by decision than those by submission.
Miarka et al. [96]	Comparison of time–motion and technical–tactical analysis between rounds.	The percentage of low-intensity activity was more prevalent in the fifth round, while the percentage of high-intensity activities was more prevalent in the fourth round. In the first and fourth rounds, strikes attempts were less frequent than in the first and second rounds.
Miarka et al. [85]	Comparison of home advantage effect on time–motion and technical–tactical analysis	Total strikes landed and attempted and head and body strike attempts were lower when the athletes were in home advantage.

**Table 2 sports-10-00080-t002:** Mean ± SD of testosterone and cortisol values in MMA fighters at selected moments.

Testosterone (nmol·L^−1^)				
	(−24 h)	(−1 h)	0 h	+24 h
Winners	20.05 ± 2.13 *	15.90 ± 1.18 *	11.92 ± 1.79 *	17.75 ± 2.00 *
Losers	15.91 ± 2.40 *	10.16 ± 3.52 *	7.14 ± 2.73 *	11.61 ± 1.79 *
Total	17.98 ± 3.06 *	13.03 ± 3.90 *	9.53 ± 3.33 *	14.68 ± 4.02 *
Cortisol (nmol·L^−1^)				
Winners	580.63 ± 83.18 *	706.00 ± 64.65 *	949.98 ± 59.20 *	522.22 ± 73.69 *
Losers	387.29 ± 147.64 Υ	482.03 ± 180.55 *	802.21 ± 94.43 *	351.69 ± 109.86 Υ
Total	483.96 ± 153.10 *	594.02 ± 144.99 *	876.09 ± 107.84 *	436.96 ± 126.26 *
Testosterone/Cortisol				
Winners	34.76 ± 2.38 Υ	22.56 ± 8.52 *	12.49 ± 1.10 *	34.13 ± 1.43 Υ
Losers	40.15 ± 4.88 *	22.85 ± 7.70 *	8.66 ± 2.42 *	10.57 ± 2.69 *
Total	37.45 ± 4.65 *	22.70 ± 5.31 *	10.57 ± 2.69 *	34.30 ± 4.73 *

Table provided with authorship by Souza et al. [104]: * *p* < 0.05 compared with other moments; Υ *p* < 0.05 compared with other moments with the exception of −24 h and +24 h; *p* < 0.05 between losers and winners.

**Table 3 sports-10-00080-t003:** Mean ± SD of glucose, lactate and creatine kinase values in MMA fighters at selected moments.

Glucose (mg/dl)				
	(−24 h)	(−1 h)	0 h	+24 h
Winners	4.48 ± 0.35 *	5.43 ± 0.83 *	11.73 ± 2.30 *	4.03 ± 0.21 *
Losers	3.80 ± 0.44 Υ	4.20 ± 0.49 *	8.27 ± 1.33 *¶	3.62 ± 0.18 *
Total	4.14 ± 0.52 *	4.81 ± 0.92 *	10.00 ± 2.55 *	3.82 ± 0.28 *
Lactate (nmol·L^−^^1^)				
Winners	1.32 ± 0.47 *	2.02 ± 0.61 Υ	2.37 ± 3.18 *	1.72 ± 0.68 Υ
Losers	1.44 ± 0.27 *	2.22 ± 0.42 *	3.88 ± 1.30 *	1.82 ± 0.47 *
Total	1.38 ± 0.38 *	2.12 ± 0.52 *	13.13 ± 2.51 *	1.77 ± 0.52 *
Creatine Kinase (U/L)				
Winners	510.17 ± 288.12 Φ	418.31 ± 277.67 +	491.40 ± 278.08 +	1304.73 ± 904.13 *
Losers	553.13 ± 128.31 ?	449.46 ± 144.27 *	492.21 ± 155.19 +	1520.48 ± 609.66 *
Total	531.15 ± 218.19 *	433.89 ± 215.95 *	491.81 ± 219.17 +	1412.69 ± 758.63 *

Table provided with authorship by Souza et al. [104]: * *p* < 0.05 compared with other moments; Υ *p* < 0.05 compared with other moments with the exception of −1 h and +24 h; Φ *p* < 0.05 compared with other moments with the exception of −1 h and 0 h; + *p* < 0.05 compared with other moments with the exception of −24 h; ? *p* < 0.05 compared with other moments with the exception of 0 h; *p* < 0001 between losers and winners. ¶ *p* < 0.05 between Losers and Winner.

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
