# Peer review of "Exploratory Systematic Review of Mixed Martial Arts: An Overview of Performance of Importance Factors with over 20,000 Athletes"

_sports, 2022, doi:10.3390/sports10060080_

Round 1

Reviewer 1 Report

Review: Systematic review of Mixed Martial Arts: an overview with 2 over 20,000 athletes

  1. Title: Systematic review of Mixed Martial Arts: an overview with 2 over 20,000 athletes

The title is not clarifying clearly the research question .An alternative title proposed can be : Systematic review of performance of importance factors in Mixed Martial Arts

  1. Abstract: Therefore, this systematic review presents to practitioners and researchers with a 14 broad summary of each facet of performance of importance in this population of athletes.

This statement should be interpreted at the tile

  1. Keywords: should not exceed more than 5 key words.

  1. Page 2 .Line 61 Therefore, the objective of the present study was to analyze the findings of the scientific literature related to MMA through a systematic review on the subject and to present the state of the art of the sport from a multifactorial perspective.

The authors must clearly define the objective of this review .Is the review exploring performance? Injuries?, epidemiology?.Must be clearly stated both in the title and the objectives of the review

  1. Page 3, Line 80: Language, search in English, must be stated as one of the limitations

  1. Page 5, 3.2.1 Injuries : The authors can have a look and consider including it at the references the following publications

 Prevalence of sport injuries in Olympic combat sports. A cross-sectional study examining one Olympic period.  Lambert C, Ritzmann R, Lambert S, Lachmann D, Malliaropoulos NG, GEßLEIN M, Peters N, Shafizadeh S. J Sports Med Phys Fitness. 2022 Feb 18. doi:

Rotational head acceleration and traumatic brain injury in combat sports: a systematic review.

Lota KS, Malliaropoulos N, Blach W, Kamitani T, Ikumi A, Korakakis V, Maffulli N. Br Med Bull. 2022 Mar 21;141(1):33-46. doi: 10.1093/bmb/ldac002.

  1. Page 8, 3.2.2 Weight loss: The authors can have a look and consider including it at the references the following publication

Prevalence, techniques and knowledge of rapid weight loss amongst adult british judo athletes: a questionnaire based study.

Malliaropoulos N, Rachid S, Korakakis V, Fraser SA, Bikos G, Maffulli N, Angioi M. Muscles Ligaments Tendons J. 2018 Jan 10;7(3):459-466.

  1. General Comments

That’s a very interesting review.

The title of the review doesn’t clarifying clearly the research question.

A paragraph on Limitations of the study must be added

What is the Bias assessment criteria of the studies included at this review?   

Author Response

REVIEWER #1:

  1. Title: Systematic review of Mixed Martial Arts: an overview with 2 over 20,000 athletes

The title is not clarifying clearly the research question. An alternative title proposed can be : Systematic review of performance of importance factors in Mixed Martial Arts

 R: The title has been changed as per your suggestions: “Exploratory systematic review of Mixed Martial Arts: an overview of performance of importance factors with over 20,000 athletes.

  1. Abstract: Therefore, this systematic review presents to practitioners and researchers with a 14 broad summary of each facet of performance of importance in this population of athletes.

This statement should be interpreted at the tile

R: This statement was interpreted at the tile and the following text was inserted in the abstract: “Therefore, this exploratory systematic review presents to practitioners and researchers with a 7 broad summary of each facet of performance of importance in this population of athletes.”

  1. Keywords: should not exceed more than 5 key words.

R: These keywords were removed: physiological; training; interventions; physical fitness.

  1. Page 2 .Line 61 Therefore, the objective of the present study was to analyze the findings of the scientific literature related to MMA through a systematic review on the subject and to present the state of the art of the sport from a multifactorial perspective. The authors must clearly define the objective of this review .Is the review exploring performance? Injuries?, epidemiology?. Must be clearly stated both in the title and the objectives of the review

R: The title, objective and conclusion were changed accordingly.

  1. Page 3, Line 80: Language, search in English, must be stated as one of the limitations

R: Thank you, we are in agreement. It was inserted in the Conclusion.

“Furthermore, according to the inclusion criteria, only manuscripts written in English were selected and this can be considered a limitation of the present study.” 

  1. Page 5, 3.2.1 Injuries : The authors can have a look and consider including it at the references the following publications

 Prevalence of sport injuries in Olympic combat sports. A cross-sectional study examining one Olympic period.  Lambert C, Ritzmann R, Lambert S, Lachmann D, Malliaropoulos NG, GEßLEIN M, Peters N, Shafizadeh S. J Sports Med Phys Fitness. 2022 Feb 18. doi:

 R: inserted in line 153: values that are lower than those reported in Olympic combat sports during training (58%) compared to with the competition (42%) [35], still on MMA,

Rotational head acceleration and traumatic brain injury in combat sports: a systematic review.

Lota KS, Malliaropoulos N, Blach W, Kamitani T, Ikumi A, Korakakis V, Maffulli N. Br Med Bull. 2022 Mar 21;141(1):33-46. doi: 10.1093/bmb/ldac002.

 R: inserted in line 195: Consequently, a possible increase in rotational acceleration (RA) is a strong predictor for traumatic brain injury (TBI) [40]. Lota et al., [40] reported that rotational acceleration (RA) is greater after direct attacks to the head. 

  1. Page 8, 3.2.2 Weight loss: The authors can have a look and consider including it at the references the following publication

Prevalence, techniques and knowledge of rapid weight loss amongst adult british judo athletes: a questionnaire based study.

Malliaropoulos N, Rachid S, Korakakis V, Fraser SA, Bikos G, Maffulli N, Angioi M. Muscles Ligaments Tendons J. 2018 Jan 10;7(3):459-466.

 R: inserted in line 274: In contrast to grappling modalities, such as judo, Malliaropoulos et al., [60] reported in their study with British judo athletes a prevalence of RWL of 84%.

  1. General Comments

That’s a very interesting review.

The title of the review doesn’t clarifying clearly the research question.

A paragraph on Limitations of the study must be added

 R: “(TableS1: Risk of bias assessment criteria) based on the study by Saw et al. [25], which has been used in previous systematic reviews [26,27].”

Reviewer 2 Report

Dear Authors:

I enjoyed reading this article very much. I found it sufficiently justified, well structured, following the PRISMA guidelines and performing a quality analysis of the articles that passed the inclusion criteria.

I only have one question that the authors should improve: they should include a more complete table with the articles that have passed the inclusion criteria where, in addition to the reference, the methodology used, the sample, the variables and whether the effect size has been calculated.

Once this issue has been resolved, I consider that the article would be ready for publication.

Yours sincerely.

Author Response

Journal: Sports

Manuscript ID: sports-1670714 (sports-1357071)

Type: Systematic Review

Dear Editor Miss Mara Pop, 

We would like to thank you and the reviewers for their kind comments on our paper. We have addressed the points raised and we would like to re-submit the revised version of the manuscript "Exploratory systematic review of Mixed Martial Arts: an overview of performance of importance factors with over 20,000 athletes". An item-by-item response is presented below. All changes in the manuscript are highlighted in red color. We appreciate the contributions from these set of reviews and we believe these amendments have improved the quality of our paper.

Sincerely yours,

João Carlos Alves and co-Authors

REVIEWER #2:

Dear Authors:

I enjoyed reading this article very much. I found it sufficiently justified, well structured, following the PRISMA guidelines and performing a quality analysis of the articles that passed the inclusion criteria.

I only have one question that the authors should improve: they should include a more complete table with the articles that have passed the inclusion criteria where, in addition to the reference, the methodology used, the sample, the variables and whether the effect size has been calculated.

Once this issue has been resolved, I consider that the article would be ready for publication.

R: Thanks for the suggestion. A table has been added that includes this additional information requested by the reviewer. It was added as Table S4. Summary of the studies

Round 2

Reviewer 1 Report

Many Thanks to the Authors for adressing the raised comments